# Error Similarity Analysis and Error Compensation of Industrial Robots with Uncertainties of TCP Calibration

**Yufei Li** [1], **Bo Li** [1,2,*], **Xidong Zhao** [3], **Simiao Cheng** [4], **Wei Zhang** [1] **and Wei Tian** [1,*]

1   College of Mechanical and Electrical Engineering, Nanjing University of Aeronautics and Astronautics, Nanjing 210016, China
2   Jiangsu Key Laboratory of Precision and Micro-Manufacturing Technology, Nanjing University of Aeronautics and Astronautics, Nanjing 210016, China
3   Academy of China Changfeng ELectro-Mechanical Technology, Beijing 100854, China
4   Hubei Institute of Aerospacecraft, Wuhan 430023, China
*   Correspondence: libo_nj@nuaa.edu.cn (B.L.); tw_nj@nuaa.edu.cn (W.T.)

**Abstract:** The machining system based on an industrial robot is a new type of equipment to meet the requirements of high quality, high efficiency and high flexibility for large and complex components of aircraft and spacecraft. The error compensation technology is widely used in robotic machining to improve the positioning accuracy of an industrial robot with the intention of meeting the precision requirements of aerospace manufacturing. However, the robot's positioning accuracy decreases significantly when the orientation of the tool changes dramatically. This stems from the fact that the existing robot compensation methods ignore the uncertainties of Tool Center Point (TCP) calibration. This paper presents a novel regionalized compensation method for improving the positioning accuracy of the robot with calibration uncertainties and large orientation variation of the TCP. The method is experimentally validated through the drilling of curved surface parts of plexiglass using a KUKA KR2830MT robot. Compared with a published error compensation method, the proposed approach improves the positioning accuracy of the robot under the large orientation variation to 0.235 mm. This research can broaden the field of robot calibration technology and further improve the adaptability of robotic machining.

**Keywords:** aircraft assembly; robot control; orientation variation; robotic machining

## 1. Introduction

Recently, robots have been increasingly used in the field of mechanical machining [1,2], especially in scenarios that require high precision, high efficiency, and flexibility, such as for aerospace assembly [3,4] and metal and composite material milling [5–7]. However, the typical positioning accuracy of the robot is merely ±1 mm [8] with the control method in the original controller provided by the robot company, which is difficult to satisfy the needs of high-precision machining tasks. Robot error compensation technology is an effective measure to cope with this problem.

Error compensation refers to the technology using measuring equipment combined with certain numerical algorithms to reduce or eliminate the positioning error of an industrial robot. Robot error compensation can be categorized into two categories: online and offline compensation. The former needs to rely on a sensor such as a laser tracker to measure the actual positions of the robot's TCP online and then to compensate for the TCP error in real-time. Electroimpact Inc. [3,9] developed an accurate industrial robot by attaching the optical encoder to each joint of the robot and customizing the Siemens numerical control system with real-time compensation algorithms. Modification of the robot joints and the control system resulted in an enhancement of the positioning accuracy from ±2–4 mm to ±0.18 mm with no restriction on tool orientation. Wang [10] proposed methods to correct the path error of a robotic machining system by using a laser tracker

of three degrees of freedom and reduced the hole position errors to 0.17 mm in a robotic drilling test. Combining a six-degrees-of-freedom laser tracker and a customized industrial robot with an optical encoder and numerical control system, Schneider [11] further improved the static absolute accuracy of the robot to 0.07 mm. Instead of the laser tracker, Gharaaty [12] used an optical coordinate measure machine to correct the error of the robot and reached an accuracy of $\pm 0.05$ mm for position and $\pm 0.05°$ for orientation. Although online feedback methods can significantly amend the accuracy of industrial robots, the high dependency on precise measuring instruments during the entire operation time is hardly acceptable in complex industrial sites.

The off-line compensation methods are typically divided into kinematic calibration and non-kinematic compensation. Kinematic calibration methods identify robot kinematic parameter errors and the transformation relationship of TCP by measurement results of the Cartesian sampling procedure. Rocadas [13] mounted three High Modulus Polyester cords on the end of the robot to measure the position of the TCP and calibrate the robot. The result showed an enhancement from an initial average positioning error of 17.88 mm–1.16 mm. Gaudreault [14] proposed a calibration method using three dial gauges and the calibration balls, which reduced the positioning error to within 0.5 mm in the target workspace. Ikits [15] compared an in-contact method using a plane constraint and an off-contact method using a 3D motion tracking system, which improved the RMS errors of the robot to 0.2515 mm and 0.156 mm, respectively. Through the laser tracker with higher precision, Wu [16] proposed a calibration method based on the six-degrees-of-freedom pose measuring strategy, which requires the mounting of three reflectors at the end of the robot and reduces the max error of the robot to 0.32 mm. To avoid the transformation error of frames between the measurement device and the robot, Wang [17] calibrated the robot by measuring the distance accuracy and increased the absolute accuracy to 0.27 mm in a single direction. However, the accuracy achieved by kinematic calibration methods still failed to meet the requirements of aerospace manufacturing, which is 0.25mm for the vast majority of applications[3], because of the neglect of non-kinematic errors.

To further enhance the accuracy of industrial robots, non-kinematic compensation methods emerged. It can be further divided into compensation methods targeting specific non-kinematic error characteristics and compensation methods based on the principle of error similarity. The former mainly includes robot compliance error [18–20] and thermal drift error [21,22]. The basic principle of error similarity is that there is a correlation between the robot positioning errors in Cartesian space and joint space. Based on this principle, one can estimate the error of desired points by the error of sampled points. Commonly used estimation methods include spatial interpolation, a neural network and the Kriging method. Tian [23] proposed an inverse distance weighted interpolation method for the robotic drilling system on the moving rail. Alici [24] compared estimation functions, including Fourier polynomials and ordinary polynomials, with robot joint angles as the variables. To further improve the accuracy of non-kinematic compensation methods, a lot of methods using neural networks have been proposed. Li [25] used a neural network optimized by genetic particle swarm optimization and reduced the positioning error of the robot to 0.227 mm via a robotic drilling test. Zhao [26] developed two hidden layers of deep neural networks to estimate the error and reached an accuracy of 0.22 mm. Wang [27] combined the inverse distance weighted method and deep belief network, which led to an accuracy of 0.24 mm. Our previous research [28,29] developed error-similarity-based estimation methods in joint space to estimate the error in the entire Cartesian workspace, achieving an accuracy of 0.3 mm. However, the existing robotic drilling system is often used in wing skin assembly scenarios, as shown in Figure 1, where the robot's orientation variation is usually less than $\pm 10°$. In these cases, the method in Ref. [28] combined with other offline compensation methods ignores the uncertainties of TCP calibration and is limited to applications with small variations in TCP orientation [30]. As a result, when the tool has a large orientation variation, the positioning accuracy of the robot varies,

and the robot accuracy after error compensation still cannot meet the requirements of aerospace manufacturing.

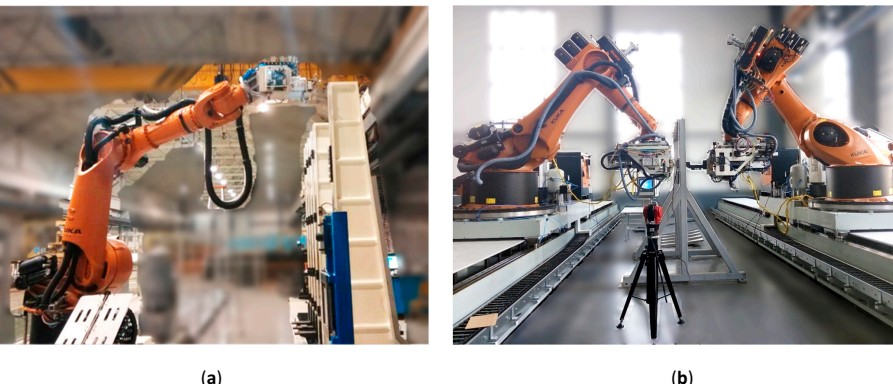

(a)                                                          (b)

**Figure 1.** Working scenarios of existing robotic drilling systems. (**a**) NUAA robotic drilling system (**b**) NUAA dual-robot cooperative drilling and riveting system.

To deal with the problem mentioned above, a regionalized robot compensation approach based on error similarity is proposed in this study. A novel TCP calibration method is proposed to reduce the initial positioning error of the robot in each region. Then, the offline compensation method based on error similarity is used to further compensate for the positioning error of the robot in each region so that the robot maintains high positioning accuracy under the large orientation variation of TCP in the whole workspace. The main contributions of this paper are listed below.

(1)    The influence of TCP calibration uncertainties on the Cartesian positioning error similarity is analyzed compared with Ref. [29], which only considered joint errors;
(2)    Considering the uncertainties of TCP calibration, a robot compensation method based on regionalized error similarity is proposed, which broadens the application limits of robot compensation technology;
(3)    The proposed method is applied successfully to the robotic drilling, which effectively reduces the positioning error under the large variation of the TCP, and the accuracy is improved from 0.96 mm with the method in Ref. [28] to 0.23 mm.

The structure of the rest of this paper is outlined as follows. In Section 2, the uncertainties in TCP calibration and their impact on the robot's error similarity are analyzed and modeled. Then, a TCP calibration method to reduce initial positioning error and a regionalized error similarity offline compensation method are presented. The experimental setup is introduced in Section 3. The experiments and their results, as well as the effectiveness of the proposed method, are discussed in Section 4. Finally, Section 5 summarizes the findings of this study.

## 2. Methods

### 2.1. Error Similarity Analysis with TCP Calibration Uncertainties

The Cartesian error similarity with respect to (w.r.t.) the joint space is analyzed in Ref. [29]. However, the uncertainties of TCP calibration are ignored and result in the limitation of applications with small variations of tool orientations. To find a compensation method that is suitable for large orientation variations of TCP, the uncertainties of TCP calibration must be taken into consideration. For this purpose, error similarity analysis with uncertainties of TCP calibration is carried out in this section.

The KUKA KR500-2830 industrial robot is considered in this study. Its DH model [31] is shown in Figure 2, where $O_i X_i Y_i Z_i$ represents the fixed frame attached to the $i$th link of the robot. In particular, $O_0 X_0 Y_0 Z_0$ represents the base frame of the robot and is denoted as {$B$}; $O_6 X_6 Y_6 Z_6$ represents the wrist frame of the robot and is denoted as {$W$}. In addition,

$O_T X_T Y_T Z_T$ represents the TCP frame of the robot and is denoted as {$T$}. The transformation matrix of TCP frame {$T$} w.r.t. base frame {$B$} can be formulated as in Ref. [13].

$$^B\boldsymbol{T}_T = {}^0\boldsymbol{T}_1\,{}^1\boldsymbol{T}_2\,{}^2\boldsymbol{T}_3\,{}^3\boldsymbol{T}_4\,{}^4\boldsymbol{T}_5\,{}^5\boldsymbol{T}_6\,{}^W\boldsymbol{T}_T \tag{1}$$

where the first 6 transformation matrices can be expressed by

$$^{i-1}\boldsymbol{T}_i = \mathrm{Trans}(0,0,d_i)\mathrm{Rot}(\theta_i,0,0)\mathrm{Trans}(a_i,0,0)\mathrm{Rot}(0,0,\alpha_i)\ i = 1,\cdots,6 \tag{2}$$

with $d_i, \theta_i, a_i, \alpha_i$ being the link offset, joint angle, link length and link twist of the $i$th link, respectively $\mathrm{Trans}(\cdot)$ and $\mathrm{Rot}(\cdot)$ represents homogeneous translations and rotations matrices which can be expressed as

$$\mathrm{Trans}(x,y,z) = \begin{bmatrix} 1 & 0 & 0 & x \\ 0 & 1 & 0 & y \\ 0 & 0 & 1 & z \\ 0 & 0 & 0 & 1 \end{bmatrix}$$
$$\mathrm{Rot}(\varphi,\theta,\psi) = \begin{bmatrix} c\varphi c\theta c\psi - s\varphi s\psi & -c\varphi c\theta s\psi - s\varphi c\psi & c\varphi s\theta & 0 \\ s\varphi c\theta c\psi + c\varphi s\psi & -s\varphi c\theta s\psi + c\varphi c\psi & s\varphi s\theta & 0 \\ -s\theta c\psi & s\theta s\psi & c\theta & 0 \\ 0 & 0 & 0 & 1 \end{bmatrix} \tag{3}$$

$^W\boldsymbol{T}_T$ in Equation (1) can be expressed by

$$^W\boldsymbol{T}_T = \mathrm{Trans}(x_T,y_T,z_T)\mathrm{Rot}(\varphi_T,\theta_T,\psi_T) \tag{4}$$

where $x_T, y_T, z_T$ represents the position of the origin for the TCP frame {$T$} w.r.t the wrist frame {$W$}. $\varphi_T, \theta_T, \psi_T$ represents the orientation transform from {$W$} to {$T$} in the *Z-Y-X* Euler angles.

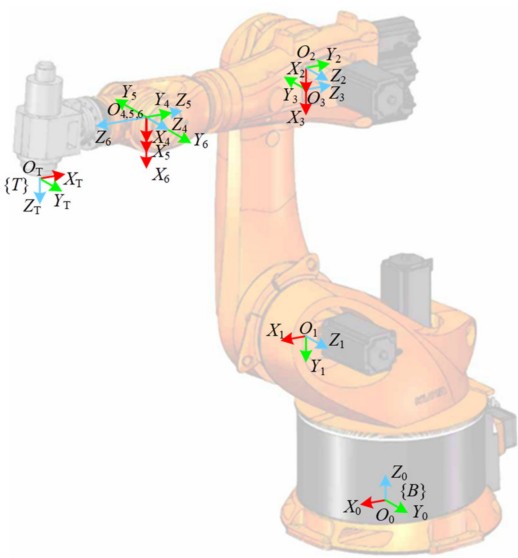

**Figure 2.** DH model of KUKA KR500-2830.

It can be known from Equation (1) that the positioning error can be divided into two parts which contain robot kinematics error $\Delta\,{}^i\boldsymbol{T}_{i+1}$ and calibration error of TCP frame $\Delta\,{}^W\boldsymbol{T}_T$. As can be seen from previous research, when the 2 positions in Cartesian space hold



similar joint configurations, the corresponding positioning errors have a certain similarity which can be calculated by semivariance [29] as

$$\gamma^*(h) = \frac{1}{2 \sum\limits_{k=1}^{n} N(h_k)} \sum_{i=1}^{n} \sum_{j=1}^{N(h_i)} \left\| \Delta \mathbf{P}(\theta_i) - \Delta \mathbf{P}(\theta_j) \right\|_2^2 \tag{5}$$

where $\|\cdot\|_2$ represents the Euclid norm of the vector. $\theta_i, \theta_j \in \mathrm{R}^6$ represent the joint angles vector of different TCP positions with a total of n groups. Particularly, for each $\theta_i$, $\theta_j \in \mathrm{R}^6$ represents the particular joint angle vector for which the Euclidean distance from $\theta_i$ is $h$. For each $\theta_i$, the total amount of corresponding $\theta_j$ is denoted as $N(h_i)$. The Euclidean distance h between two joint configurations can be expressed by Euclid norm as

$$h = \left\| \theta_i - \theta_j \right\|_2 \tag{6}$$

$\Delta P(\theta_i)$, $\Delta P(\theta_j) \in \mathrm{R}^3$ represents the TCP positioning errors of the corresponding joint configurations and takes the form of

$$\Delta P(\theta_\ell) = P(\theta_\ell) - (P(\theta_\ell + \Delta \theta)) \; \ell = i, \; j \tag{7}$$

where $P(\theta_\ell)$ represents the theoretical position of TCP with the joint configuration of $\theta_\ell$. $\Delta \theta$ is the deviation of joint angles.

However, Equation (7) holds only under the assumption that there is no error in calibrating the TCP frame. When the error exists, Equation (7) must be expanded to the following homogeneous coordinates form.

$$\begin{bmatrix} \Delta P'(\theta_\ell) \\ 1 \end{bmatrix} = \left( \begin{bmatrix} P(\theta_\ell) \\ 1 \end{bmatrix} - \begin{bmatrix} P(\theta_\ell + \Delta \theta) \\ 1 \end{bmatrix} \cdot \Delta^W T_T \right) \cdot \begin{bmatrix} O_{3\times 1} \\ 1 \end{bmatrix} \tag{8}$$

where $\Delta^W T_T$ is the calibration error of the TCP frame. $O_{3\times 1}$ represents the 0-column vector in 3 dimensions.

Considering Equation (2), because the only variable is $\theta_i$. $\mathrm{Trans}(0,0,d_i)$, $\mathrm{Trans}(a_i,0,0)$ and $\mathrm{Rot}(0,0,\alpha_i)$ in Equation (2) are all constant matrices. In order to simplify the operation, it can be assumed that

$$\begin{aligned} d_i &= a_i = \alpha_i = 0, \; i = 1, \cdots, 6 \\ \Delta \theta &= [\Delta \theta_1, \cdots, \Delta \theta_6] \end{aligned} \tag{9}$$

which leads to $\mathrm{Trans}(0,0,d_i)$, $\mathrm{Trans}(a_i,0,0)$ and $\mathrm{Rot}(0,0,\alpha_i)$ transforming into the identity matrix $I_{4\times 4}$.

In addition, since the TCP frame calibration error $\Delta^W T_T$ is independent of $^W T_T$, it can be assumed that

$$\begin{aligned} ^W T_T &= \begin{bmatrix} I_{3\times 3} & O_{3\times 1} \\ O_{1\times 3} & 1 \end{bmatrix} \\ \Delta^W T_T &= \mathrm{Trans}(\Delta x_T, \Delta y_T, \Delta z_T) \mathrm{Rot}(\Delta \varphi_T, \Delta \theta_T, \Delta \psi_T) \end{aligned} \tag{10}$$

where $\Delta x_T, \Delta y_T, \Delta z_T$ are the translation errors, and $\Delta \varphi_T, \Delta \theta_T, \Delta \psi_T$ are the rotation error.

Substituting Equations (9) and (10) into Equation (7), Equation (7) can be rewritten as

$$\begin{bmatrix} \Delta P(\theta_\ell) \\ 1 \end{bmatrix} = \left( \prod_{k=1}^{6} \mathrm{Rot}(\theta_{\ell,k}, 0, 0) - \prod_{k=1}^{6} \mathrm{Rot}(\theta_{\ell,k} + \Delta \theta_k, 0, 0) \right) \cdot \begin{bmatrix} O_{3\times 1} \\ 1 \end{bmatrix} \; \ell = i, \; j \tag{11}$$

where $\theta_{\ell,k}$ and $\Delta\theta_k$ are the elements of $\theta_\ell$ and $\Delta\theta$ respectively. Similarly, by substituting Equation (9) and Equation (10) into Equation (8), Equation (8) can be rewritten as

$$
\begin{bmatrix} \Delta P^{'}(\theta_\ell) \\ 1 \end{bmatrix} = \left( \prod_{k=1}^{6} \mathrm{Rot}(\theta_{\ell,k},0,0) - \left( \prod_{k=1}^{6} \mathrm{Rot}(\theta_{\ell,k}+\Delta\theta_k,0,0) \right) \Delta^{\mathrm{W}} T_{\mathrm{T}} \right) \cdot \begin{bmatrix} O_{3\times 1} \\ 1 \end{bmatrix} \quad \ell = i,\, j
$$

$$(12)$$

By combining Equations (5), (11) and (12), one can obtain

$$
\gamma^*(h) - \hat{\gamma}^*(h) = \frac{1}{2 \sum\limits_{k=1}^{n} N(h_k)} \sum_{i=1}^{n} \sum_{j=1}^{N(h_i)} \| \left( 2\Xi_i - \left( \Delta^{\mathrm{W}} T_{\mathrm{T}} + I_{4\times 4} \right) \Delta\Xi_i \right) \left( \left( \Delta^{\mathrm{W}} T_{\mathrm{T}} - I_{4\times 4} \right) \Delta\Xi_i \right)
$$

$$
+ \left( 2\Xi_j - \left( \Delta^{\mathrm{W}} T_{\mathrm{T}} + I_{4\times 4} \right) \Delta\Xi_j \right) \left( \left( \Delta^{\mathrm{W}} T_{\mathrm{T}} - I_{4\times 4} \right) \Delta\Xi_j \right)
$$

$$
+ 2 \left( \Delta\Xi_i \left( I_{4\times 4} - \Delta^{\mathrm{W}} T_{\mathrm{T}} \right) \Xi_j + \Xi_i \Xi_j \left( I_{4\times 4} - \Delta^{\mathrm{W}} T_{\mathrm{T}} \right) \right.
$$

$$
+ \left( \Delta\Xi_i \Delta^{\mathrm{W}} T_{\mathrm{T}} \Delta\Xi_j \right) \left( \Delta^{\mathrm{W}} T_{\mathrm{T}} - I_{4\times 4} \right) + \Delta\Xi_i \left( \Delta^{\mathrm{W}} T_{\mathrm{T}} - I_{4\times 4} \right) \Delta\Xi_j ) \Big] \begin{bmatrix} O_{3\times 1} \\ 1 \end{bmatrix} \|_2^2
$$

$$(13)$$

where

$$
\Xi_\ell = \prod_{k=1}^{6} \mathrm{Rot}(\theta_{\ell,k},0,0)
$$
$$
\Delta\Xi_\ell = \prod_{k=1}^{6} \mathrm{Rot}(\theta_{\ell,k}+\Delta\theta_k,0,0) \qquad \ell = i,\, j
$$

$$(14)$$

In Equation (13), $\gamma^*(h)$ and $\hat{\gamma}^*(h)$ represent the situation that the semivariances with and without TCP frame calibration error, respectively. When TCP frame calibration error is ignored,

$$
\Delta^{\mathrm{W}} T_{\mathrm{T}} - I_{4\times 4} = O_{4\times 4}
$$

$$(15)$$

However, due to the calibration error, Equation (15) cannot hold strictly, which leads to Equation (13) not being equal to 0. Therefore, the similarity of TCP positioning error is affected by TCP calibration error.

According to Equation (8), 500 sets of robot positioning errors are randomly generated, and the random errors of each joint angle of the robot are set to a maximum of $\pm 0.1°$. The calibration error of the TCP frame in each direction is set to a maximum of 0 mm, $\pm 1$ mm and $\pm 5$ mm, respectively. The similarity between the positioning errors and the Euclidean distances of the robot joints is shown in Figures 3, 4 and 10. In the figures, the horizontal axis represents the Euclidean distance between corresponding joints in 2 different robot configurations, while the vertical axis represents the deviation of the positioning error between the two corresponding robot configurations.

From Figure 3, it can be seen that without considering the TCP frame establishment error, there is a correlation between the positioning errors of any two points in the Cartesian space and the distance in the joint space. That is, the smaller the distance in joint space (the horizontal axis in the figure), the smaller the difference in positioning errors in the Cartesian space (the vertical axis). Therefore, the unbiased optimal estimation can be used to estimate positioning errors in the Cartesian space.

However, when TCP frame establishment errors are introduced, joint angle errors are coupled with the establishment error. More specifically, affected by TCP calibration error, the similarity in the z-direction seriously loses when the max calibration error is $\pm 1$ mm, and so as the x- and y-directions when the max error increase to $\pm 5$ mm. Additionally, as the calibration error increases to $\pm 1$ mm and further to $\pm 5$ mm, the deviation of pose error rises rapidly by more than tenfold. This indicates that estimating the positioning errors in Cartesian space using nearby points in joint space not only fails to effectively compensate for the positioning errors but may even amplify them, resulting in larger errors, as shown in Figures 4 and 10. To sum up, even small TCP calibration errors can result in

great impacts on the similarity and predictability of the positioning error. Therefore, it is necessary to eliminate TCP calibration errors. Meanwhile, error similarity varies greatly in different directions, which means that the error estimation model needs to be distinct in each direction.

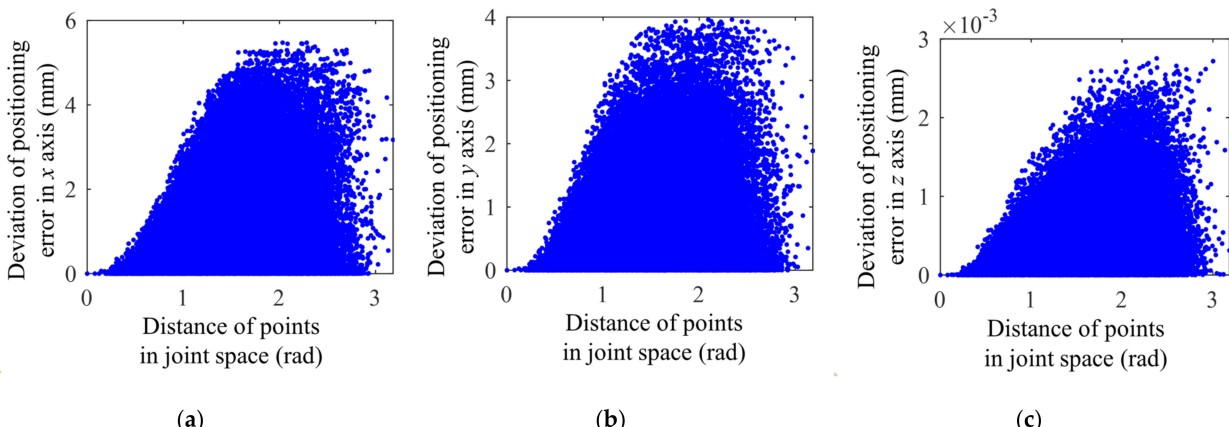

**Figure 3.** Error similarity analysis without TCP calibration error. (**a**) Similarity in the x-axis; (**b**) similarity in the y-axis; (**c**) similarity in the z-axis.

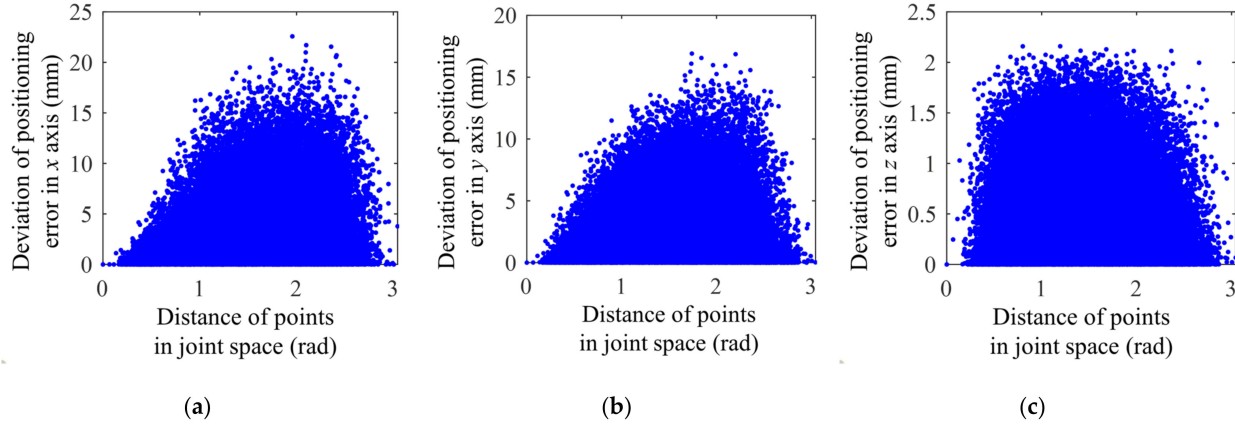

**Figure 4.** Error similarity analysis with TCP calibration error within ±1 mm. (**a**) Similarity in the x-axis; (**b**) similarity in the y-axis; (**c**) similarity in the z-axis.

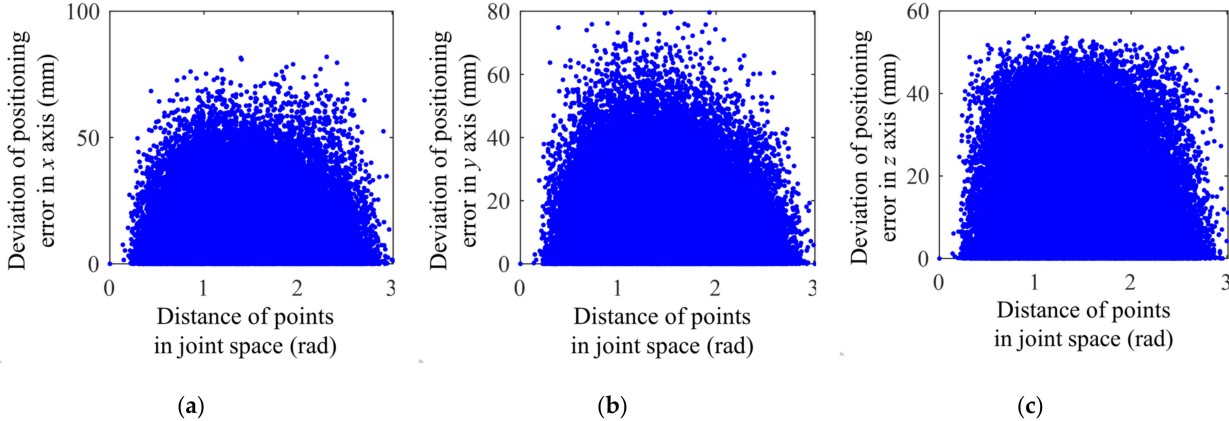

**Figure 5.** Error similarity analysis with TCP calibration error within ±5 mm (**a**) Similarity in the x-axis; (**b**) similarity in the y-axis; (**c**) similarity in the z-axis.

### 2.2. Regionalized Calibration of TCP

Since the calibration error of frames is hard to eliminate, we use {BR}, {WR} and {TR} to represent the actual base, wrist and TCP frames, which can be fitted by measuring apparatus. {B}, {W} and {T} represent the theoretical frames of the robot, which cannot be obtained by measurement. The transformation relationships between the frames are represented as

$$
\begin{aligned}
{}^{\text{World}}T_{\text{BR}} \cdot {}^{\text{BR}}T_{\text{WR}} \cdot {}^{\text{WR}}T_{\text{TR}} &= {}^{\text{World}}T_{\text{TR}} \\
{}^{\text{World}}T_{\text{B}} \cdot {}^{\text{B}}T_{\text{W}} \cdot {}^{\text{W}}T_{\text{T}} &= {}^{\text{World}}T_{\text{T}}
\end{aligned}
\tag{16}
$$

Assuming that the desired TCP frame of the robot is {TD} when the TCP moves to the desired position, the following equation holds,

$$
{}^{\text{TR}}\mathbf{T}_{\text{TD}} = \begin{bmatrix} \mathbf{I}_{3\times3} & \mathbf{O}_{3\times1} \\ \mathbf{O}_{1\times3} & 1 \end{bmatrix}
\tag{17}
$$

In other words, should the TCP reach the desired position, {TR}, which coincides with {TD}, Equation (17) must be strictly satisfied. However, because of the calibration error of the base frame $\Delta\,{}^{\text{World}}T_{\text{B}}$, TCP frame $\Delta\,{}^{\text{W}}T_{\text{T}}$ and robot kinematics error $\Delta\,{}^{\text{B}}T_{\text{W}}$, Equation (17) is hard to hold strictly. The relationship between actual frames, theoretical frames and errors can be formulated as

$$
\begin{aligned}
{}^{\text{World}}T_{\text{B}} \cdot \Delta\,{}^{\text{World}}T_{\text{B}} &= {}^{\text{World}}T_{\text{BR}} \\
{}^{\text{B}}T_{\text{W}} \cdot \Delta\,{}^{\text{B}}T_{\text{W}} &= {}^{\text{BR}}T_{\text{WR}} \\
{}^{\text{B}}T_{\text{W}} \cdot \Delta\,{}^{\text{W}}T_{\text{T}} &= {}^{\text{WR}}T_{\text{TR}}
\end{aligned}
\tag{18}
$$

Substituting Equation (18) into Equation (16), one can obtain

$$
{}^{\text{World}}T_{\text{B}} \cdot \Delta\,{}^{\text{World}}T_{\text{B}} \cdot {}^{\text{B}}T_{\text{W}} \cdot \Delta\,{}^{\text{B}}T_{\text{W}} \cdot {}^{\text{W}}T_{\text{T}} \cdot \Delta\,{}^{\text{W}}T_{\text{T}} = {}^{\text{World}}T_{\text{TR}}
\tag{19}
$$

As mentioned in Section 2.1, the error similarity of the robot in the whole workspace is reduced due to the existence of TCP calibration error. To solve the problem above, regionalized error similarity is proposed. The robot workspace can be divided into several regions to be calibrated. Assuming that the positioning error at the center of each region can be reduced as much as possible, the positioning error of all desired positions in this area can be reduced. In order to realize the assumption, control {TR} to make it as close to the center of the region as possible. Afterward, calibrate the {WR} by the relationship between {B} and {W}. Then calibrate {TR} at the center point denoted as {TDR}. Finally, calculate the relationship between {TDR} and {WR} by

$$
{}^{\text{World}}T_{\text{BR}} \cdot {}^{\text{B}}T_{\text{W}} \cdot {}^{\text{WR}}T_{\text{TDR}} = {}^{\text{World}}T_{\text{TDR}}
\tag{20}
$$

By combining Equations (18), (19) and (20), one can obtain

$$
\begin{aligned}
{}^{\text{World}}T_{\text{TR}} &= {}^{\text{World}}T_{\text{TDR}} \\
{}^{\text{WR}}T_{\text{TDR}} &= \Delta\,{}^{\text{B}}T_{\text{W}} \cdot {}^{\text{W}}T_{\text{T}} \cdot \Delta\,{}^{\text{W}}T_{\text{T}}
\end{aligned}
\tag{21}
$$

Equation (21) shows that when ${}^{\text{WR}}T_{\text{TDR}}$ is used as ${}^{\text{W}}T_{\text{T}}$, although actual frames {BR} and {WR} cannot coincide with theoretical frames {B} and {W}, it eliminates the kinematics error and TCP calibration uncertainties. As a result, the positioning error of TCP reduces in the whole region.

### 2.3. Regionalized Error Similarity Compensation Method

According to the character of error similarity, the positioning error of the robot in Cartesian space can be expressed as an unbiased optimal estimation model related to its joint configuration, and the error of the desired position is estimated depending on the error of the sampling points in the space. However, it can be seen from Section 2

that the TCP calibration error will significantly affect the accuracy of the error estimation. In addition, even if the error estimation method can compensate more than 90% of the positioning error, when the initial positioning error of the robot is too large due to the large orientation variation of the TCP frame, the residual error will still make its positioning accuracy difficult to meet the requirements of the aerospace manufacturing.

For the purpose of solving the above problems, a regionalized error compensation method considering TCP calibration error is proposed. The specific steps are as follows.

Step 1: divide the robot workspace into $n$ regions denoted as $\Omega_1, \Omega_2, \cdots, \Omega_n$ according to the orientation variation of desired TCP frames, as shown in Figure 6. And the center point of each region is selected as $\{TDR_1\}, \{TDR_2\}, \cdots, \{TDR_n\}$.

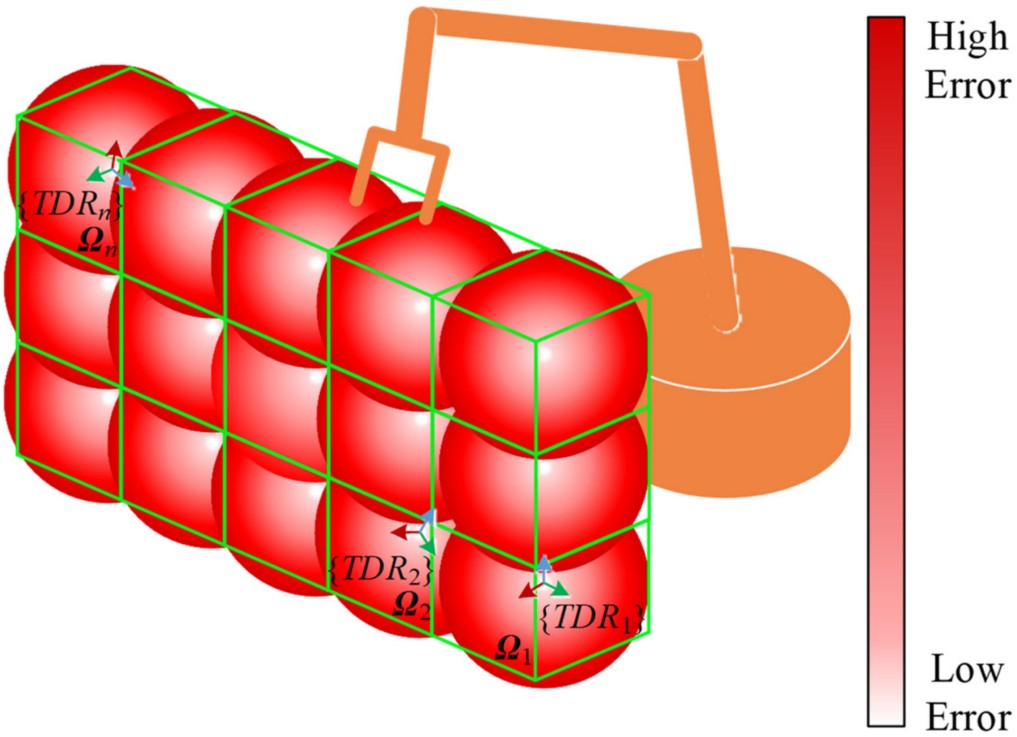

**Figure 6.** Regionalized error similarity compensation method.

Step 2: use the method of Section 2.2 to calibrate the transformation matrix from $\{WR\}$ to $\{TR\}$ in each region and denoted as $^{\mathrm{WR}}T_{\mathrm{TDR1}}, {}^{\mathrm{WR}}T_{\mathrm{TDR2}}, \cdots {}^{\mathrm{WR}}T_{\mathrm{TDR}n}$. Then measure the positioning error of the robot in each region.

Step 3: for any desired TCP frame $\{TD_x\}$, select the corresponding region $\Omega_x$ according to its orientation and solve the theoretical joint configuration $\theta_x$ according to the kinematics parameters.

Step 4: the method of Ref. [29] is used to estimate the error of the desired TCP frame.

Step 5: because the estimated error is based on the robot error sampling result, that is, the positioning error when the transformation matrix from $\{WR\}$ to $\{TR\}$ $^{\mathrm{WR}}T_{\mathrm{TDR}x}$, which is calibrated in the corresponding region. Therefore, it is necessary to transform the calibrated desired TCP frame from $^{\mathrm{WR}}T_{\mathrm{TDR}x}$ to the default $^{\mathrm{W}}T_{\mathrm{T}}$ of the robot and finally get the desired TCP frame after calibration, which is denoted as $\{TD_x\}'$.

For any kind of robot and control system, the above compensation method can be represented by the flow shown in Figure 7. The robot control system (RCS) only needs to send the desired TCP frame to the position compensation system (PCS) and receive the compensated TCP frame from PCS. Obviously, the proposed compensation method has good versatility for any kind of robot and control system.

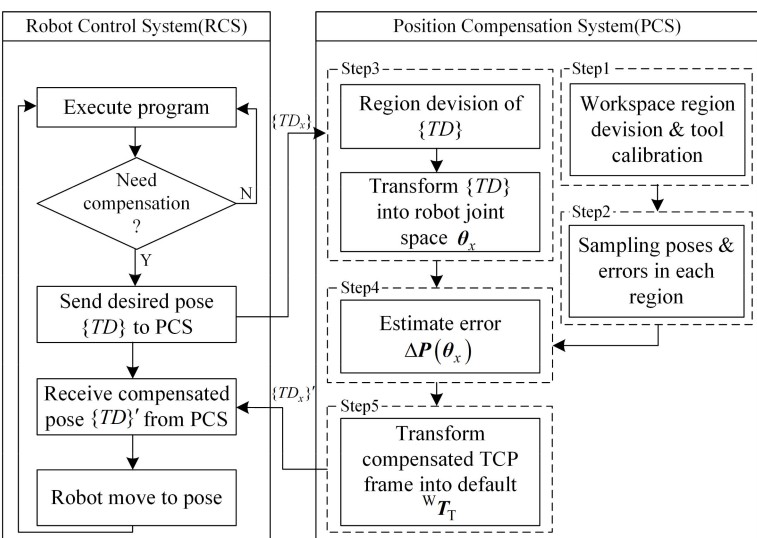

**Figure 7.** Robot error compensation process.

## 3. Experimental Studies

The experimental setup is shown in Figure 8, consisting of a KUKA KR500-2830MT industrial robot (KUKA Robotics China Co.,Ltd., Shanghai, China) controlled by a Siemens 840Dsl numerical control system (Siemens Ltd., Berlin and Munich, Germany) and a Reckerth RF170D motor spindle (Hugo Reckerth GmbH, Filderstadt-Bonlanden, Germany). The positioning errors are collected by a Faro VantageE laser tracker (FARO Technologies Inc., Lake Mary, FL, USA). In order to reduce the error caused by repetitive fitting base frame $\{BR\}$ and the possible temperature drift and zero drift produced by the measuring equipment in the sampling process, three reference targets are set on the platform, and the platform frame $\{P\}$ is calibrated through these three targets. The transformation relationship between $\{P\}$ and $\{BR\}$ is $^{P}T_{BR}$. During the regionalized TCP calibration and error sampling process, only $\{P\}$ is re-calibrated, and the actual base frame $\{BR\}$ can be obtained by $^{World}T_{BR} = {}^{World}T_{P} \cdot {}^{P}T_{BR}$.

When executing the NC program in the robot control system, auxiliary code has been added in advance by means of offline programming to indicate that a certain line of command needs to be calibrated. The user-defined HMI interface and R parameters can be used to interact with NC programs so as to transmit the desired TCP frames and compensated TCP frames. Hence, the PCS is developed by an HMI interface and communicates with the 840Dsl CNC system through Ethernet.

The drilling task of typical aeronautical curved surface parts faced by the robotic machining system is shown in Figure 9. For the purpose of verifying the regionalized compensation method proposed in this paper, the drilling area in the range of $-110°--40°$ and $40°-110°$ are divided into Region 1 ($\Omega_1$), Region 2 ($\Omega_2$), Region 3 ($\Omega_3$) and Region 4 ($\Omega_4$). The material of the parts to be processed is plexiglass. The spindle speed and feeding speed of the drilling process are 3000 rpm and 1 mm/s, respectively.

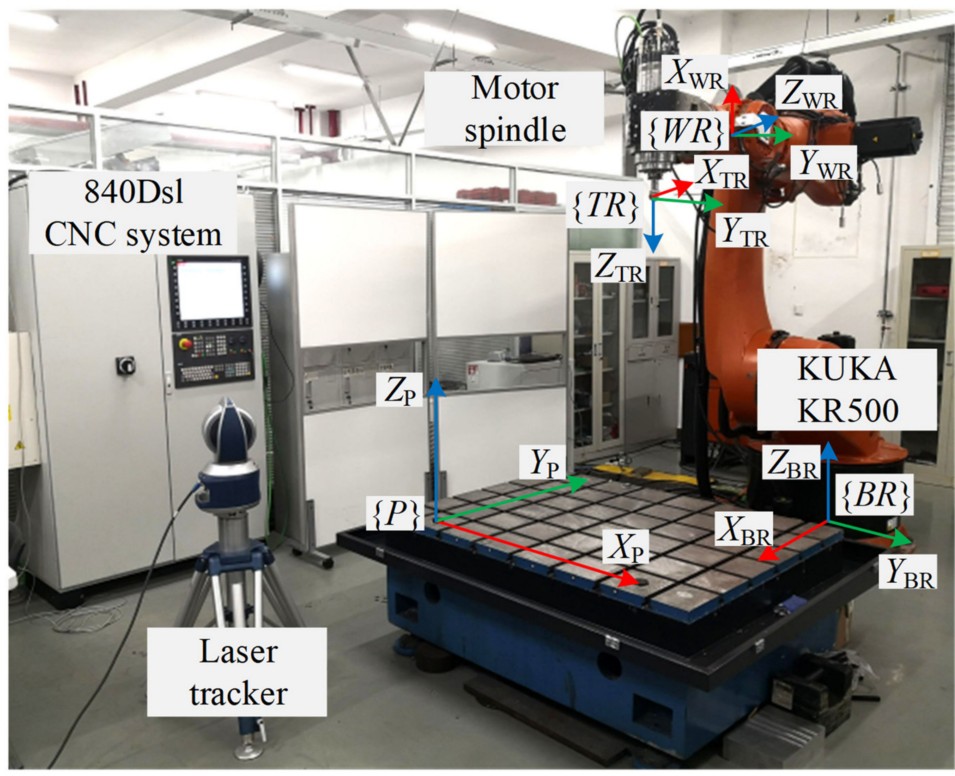

**Figure 8.** Experimental setup of the robotic drilling system.

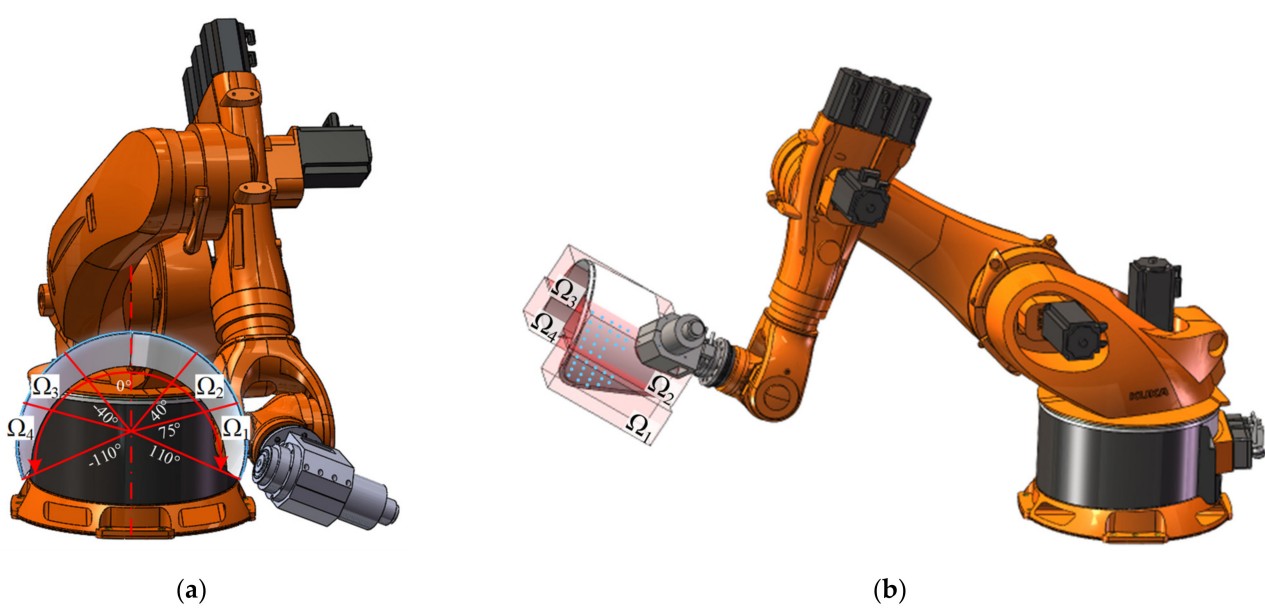

**Figure 9.** Robotic drilling task. (**a**) Region division of robotic drilling task; (**b**) Sampling area division of robotic drilling task.

## 4. Results

In order to avoid the influence of kinematics parameter errors excepting joint angles on the pose error, the kinematics parameters of the robot, including 24 DH parameters and the default $^W T_T$ are calibrated by the least squares method in Ref. [13]. The calibrated DH parameters are listed in Table 1.

**Table 1.** Calibrated DH parameters of KUKA KR500-2830.

| Joint Frame | $\theta$ (°) | $a$ (mm) | $d$ (mm) | $\alpha$ (°) |
|---|---|---|---|---|
| $\{B\}$ | 0 | 0 | 0 | 0 |
| $\{O_1 X_1 Y_1 Z_1\}$ | $\theta_1$ | 500.5065 | 1045 | −90 |
| $\{O_2 X_2 Y_2 Z_2\}$ | $\theta_2$ | −1300.4225 | 0 | 0 |
| $\{O_3 X_3 Y_3 Z_3\}$ | $\theta_3$ | 54.39 | 0 | −90 |
| $\{O_4 X_4 Y_4 Z_4\}$ | $\theta_4$ | 0 | −1025.1913 | 90 |
| $\{O_5 X_5 Y_5 Z_5\}$ | $\theta_5$ | 0 | 0 | −90 |
| $\{O_6 X_6 Y_6 Z_6\}$ | $\theta_6$ | 0 | 0 | 180 |

The $\{TDR\}$ and $^W T_T$ is calibrated by the proposed means in Section 2.2 in the center of each region. The results are shown in Table 2, where the $^W T_T$ calibrated by the least square method is denoted as $\Omega_0$.

**Table 2.** Calibration results of $^W T_T$ in different regions.

| Region | $\Omega_0$ | $\Omega_1$ | $\Omega_2$ | $\Omega_3$ | $\Omega_4$ |
|---|---|---|---|---|---|
| $x_T$ (mm) | 453.31 | 452.23 | 452.03 | 451.24 | 450.50 |
| $y_T$ (mm) | 5.21 | −4.71 | −3.68 | −3.32 | −3.52 |
| $z_T$ (mm) | 384 | 383.80 | 383.53 | 386.91 | 387.70 |
| $\varphi_T$ (°) | −140.18 | −77.64 | −57.3 | 87.55 | 128.34 |
| $\theta_T$ (°) | −89.99 | −89.91 | −89.93 | −89.86 | −89.85 |
| $\psi_T$ (°) | 140.56 | 76.9224 | 56.73 | −87.91 | −128.74 |

It can be seen that the maximum difference of the $^W T_T$ appears in the $Y$ direction, and the maximum deviation of $Y$ from $\Omega_1$ to $\Omega_4$ is 1.39 mm, but the maximum deviation of $Y$ from working regions to $\Omega_0$ is 9.92 mm, which is caused by the large change of the orientation in the $YOZ$ plane of the robot base frame. Simultaneously, it can be seen from Table 3 that the distances between the origin of four $^W T_T$ in the drilling regions from $\Omega_1$ to $\Omega_4$ (up to 4.44 mm) are much smaller than those between the drilling regions and $\Omega_0$ (up to 9.98 mm), which shows that the positioning error of the robot with large orientation variation seriously affects the positioning accuracy of the actual wrist frame, so that the $^W T_T$ calibrated by the least square method is no longer applicable. However, the distance of the ipsilateral regions ($\Omega_1$ vs. $\Omega_2$ and $\Omega_3$ vs. $\Omega_4$) is 1.08 mm and 1.1 mm, which indicates that even if the positions are close, $^W T_T$ will also be affected by the orientation variation.

**Table 3.** Calibration results of $^W T_T$ in different regions.

| Error (mm) | $\Omega_0$ | $\Omega_1$ | $\Omega_2$ | $\Omega_3$ | $\Omega_4$ |
|---|---|---|---|---|---|
| $\Omega_0$ | | 9.98 | 8.99 | 9.25 | 9.89 |
| $\Omega_1$ | 9.98 | | 1.08 | 3.55 | 4.43 |
| $\Omega_2$ | 8.99 | 1.08 | | 3.49 | 4.44 |
| $\Omega_3$ | 9.25 | 3.55 | 3.49 | | 1.1 |
| $\Omega_4$ | 9.89 | 4.43 | 4.44 | 1.1 | |

As shown in Figure 9b, according to the spatial distribution of the drilling task, the length, width and height of sampling space $\Omega_1$, $\Omega_2$, $\Omega_3$ and $\Omega_4$ are 300 mm × 300 mm × 500 mm. Latin hypercube sampling [25] is used to plan 100 sampling points in each region, and the positioning errors of {TR} with the $^W T_T$ calibrated in $\Omega_0$, $\Omega_1$, $\Omega_2$ $\Omega_3$ and $\Omega_4$ are collected respectively. After that, the positioning error of the robot is calibrated by the method proposed by Ref. [28] and this paper for comparison. The robot positioning accuracy comparison results are shown in Figure 10, where all 64 desired drilling poses in $\Omega_1$, 27 poses in $\Omega_2$, 45 poses in $\Omega_3$ and 50 poses in $\Omega_4$ are tested according to the part drawing.

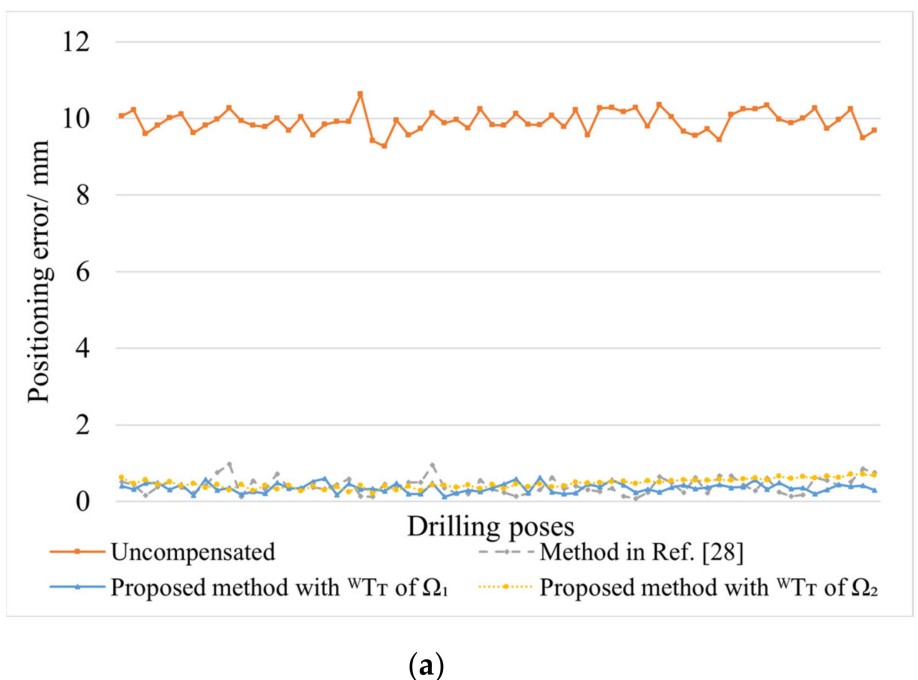

(**a**)

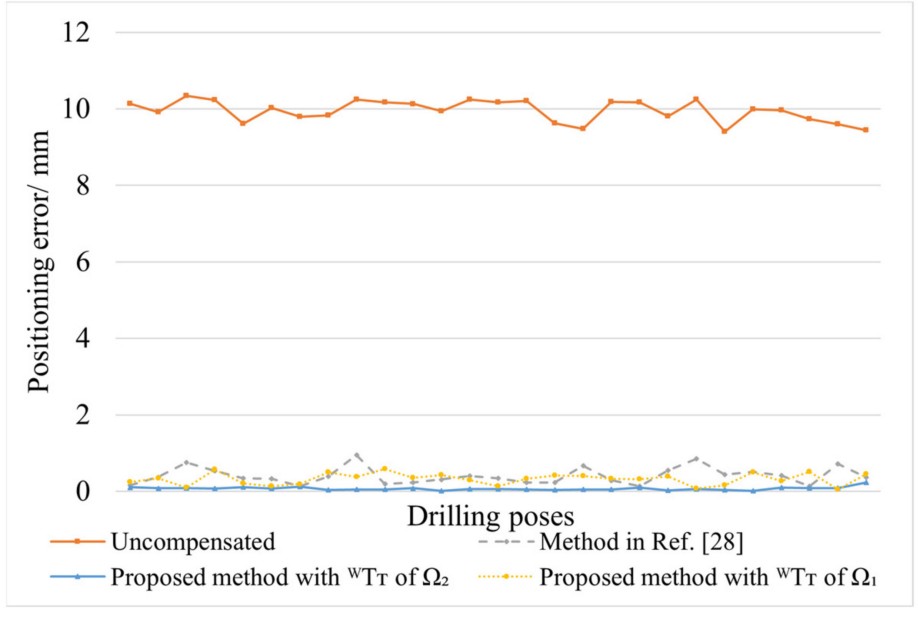

(**b**)

**Figure 10.** *Cont.*

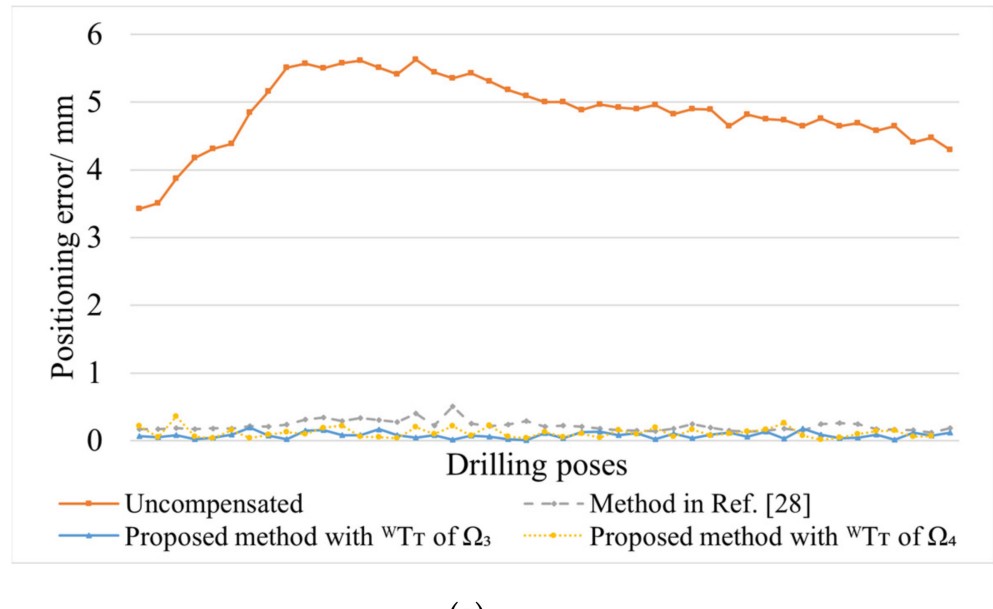

(c)

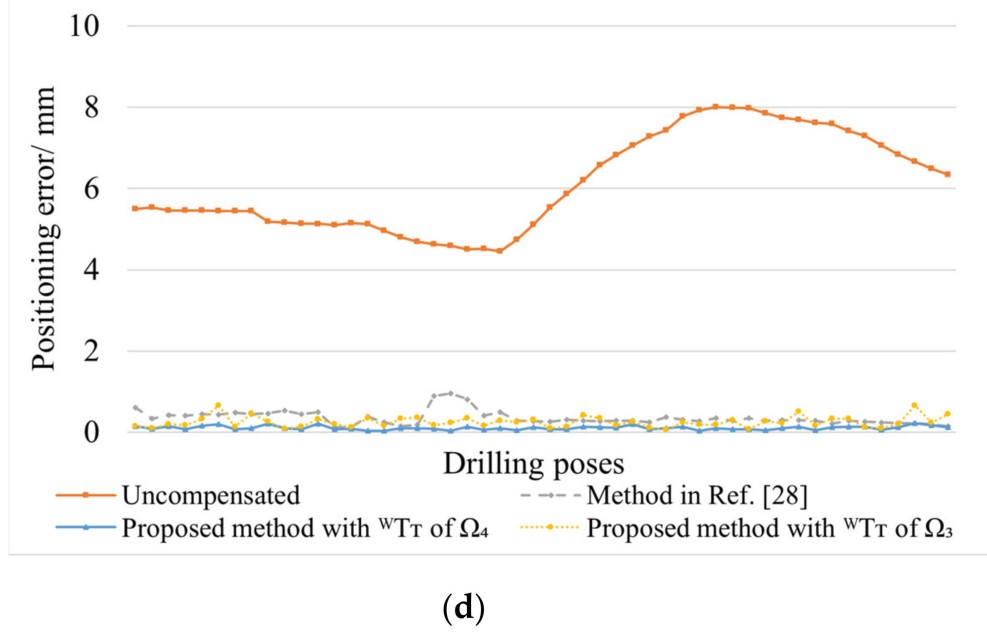

(d)

**Figure 10.** Experimental results of robot compensation. (**a**) Positioning error of $\Omega_1$ (**b**) Positioning error of $\Omega_2$; (**c**) Positioning error of $\Omega_3$; (**d**) Positioning error of $\Omega_4$.

The experimental results demonstrate that for any region, the proposed method and the method in Ref. [28] both effectively reduce the positioning error. In detail, the differences between the method in Ref. [28] and the proposed method with $^{W}T_{T}$ calibrated in the corresponding regions (blue lines vs. gray lines) are caused by the difference between compensation methods. The differences between the proposed method with $^{W}T_{T}$ calibrated in adjacent regions (blue lines vs. yellow lines) are caused by the calibration uncertainties of the TCP frame, which is merely 1.08 mm between $\Omega_1$ and $\Omega_2$ and 1.1 mm between $\Omega_3$ and $\Omega_4$ in total distance. The statistical result of the experiment is demonstrated in Table 4.

**Table 4.** Statistical result of the experiment.

| Methods | Error Range of $\Omega_1$ (mm) | Error Range of $\Omega_2$ (mm) | Error Range of $\Omega_3$ (mm) | Error Range of $\Omega_4$ (mm) |
|---|---|---|---|---|
| Uncompensated | [9.2737, 10.6342] | [9.4061, 10.3392] | [3.4253, 5.6238] | [4.455, 7.997] |
| Method in Ref. [28] | [0.0723, 0.9507] | [0.1288, 0.8524] | [0.1216, 0.5048] | [0.1194, 0.9593] |
| Proposed method with corresponding $^W T_T$ | [0.0623, 0.2099] | [0.011, 0.2348] | [0.0058, 0.1952] | [0.031, 0.2178] |
| Proposed method with $^W T_T$ of the adjacent region | [0.2114, 0.7061] | [0.0542, 0.5798] | [0.0162, 0.3565] | [0.0602, 0.6618] |

As seen in the results, the method in Ref. [28] is able to effectively reduce the maximum error in $\Omega_1$ from 10.6342 mm to 0.9507 mm, resulting in an improvement of accuracy by 91.06%. Similarly, the accuracy in $\Omega_2$, $\Omega_3$ and $\Omega_4$ increased by 91.76%, 91.02% and 88%, respectively. Despite the significant improvement in error compensation of about 90% in each region, it still falls short of meeting the requirements of a drilling accuracy of 0.25 mm.

In comparison, when using the method proposed in this paper with $^W T_T$ calibrated in the corresponding region, the maximum errors in $\Omega_1$, $\Omega_2$, $\Omega_3$ and $\Omega_4$ are reduced to 0.21 mm, 0.23 mm, 0.2 mm and 0.22 mm, respectively. As opposed to the method in Ref. [28], the accuracy is further improved by 77.91%, 75.08%, 61.33% and 77.3%, which meets the accuracy requirements of the drilling task. However, when the TCP frames are calibrated in the adjacent region, the maximum errors in $\Omega_1$, $\Omega_2$, $\Omega_3$ and $\Omega_4$ rise to 0.7061 mm, 0.5798 mm, 0.3565 mm and 0.6618 mm, proving that the division strategy of the region is reasonable.

Other scientists also developed different compensation methods for robots with a large orientation variation. The comparison can be seen in Table 5. In reference [30], Cao divided the joint space into two 3-dimensional subspaces and created spatial grids to compensate for both positioning and orientation errors. However, the positioning accuracy after compensation was still 0.334 mm, which did not meet the requirements for the drilling task. On the other hand, Electroimpact [3] developed high-accuracy robots with an accuracy of 0.18 mm by using optical encoders on each joint and integrating a real-time compensation algorithm into the CNC system. However, this method has the limitations of requiring hardware modification and the real-time compensation algorithm being only accessible to a limited group of companies. In comparison, the method proposed in this paper offers a high accuracy compensation method that is suitable for all robots.

**Table 5.** Comparison with other compensation methods considering orientation variation.

| Methods | Error without Compensation (mm) | Error after Compensation (mm) | Orientation Variation (°) | Secondary Encoder Needed (Y/N) | Number of Sampling Points |
|---|---|---|---|---|---|
| Electroimpact [3] | Not available | 0.18 | No restriction | Y | 600 |
| Cao [30] | 8.473 | 0.334 | [−20°, 20°] | N | 750 |
| This paper | 10.6342 | 0.2348 | [−110°, 110°] | N | 400 |

With the aim of further studying the similarity characteristics of positioning errors, the spatial distances and orientation deviations between each desired drilling position and {TDR} in $\Omega_1$ and $\Omega_2$ are calculated, and the regression line is fitted, as shown in Figure 11. Through the comparison of regression functions in Figure 11a,b, it is apparent that in $\Omega_1$, the effect of orientation on error similarity is significantly higher than that of spatial distance; that is, the more similar the orientation is, the smaller the residual error is. On the other hand, Figure 11c shows that in $\Omega_2$, the residual error decreases with the increase of spatial distance, which indicates that the similarity of error does not increase with the decrease of distance in Cartesian space, but the accuracy enhances with the descends of orientation deviation, which further proves that the influence of orientation deviation on robot positioning error is greater than that of spatial distance.

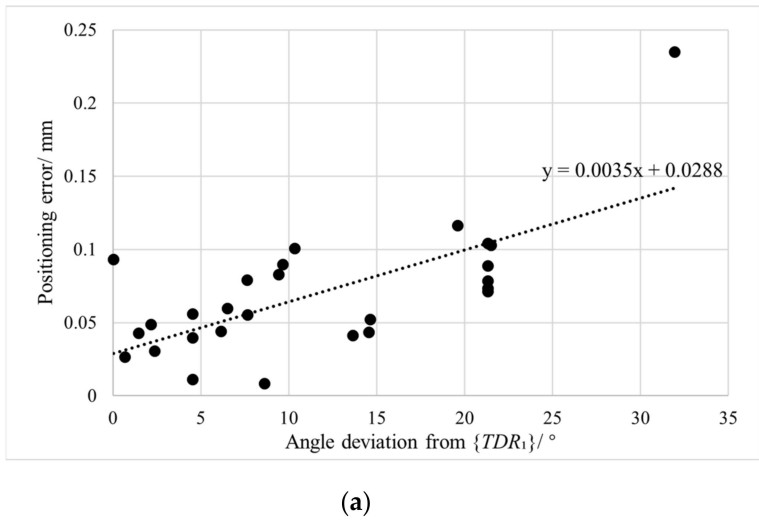

(**a**)

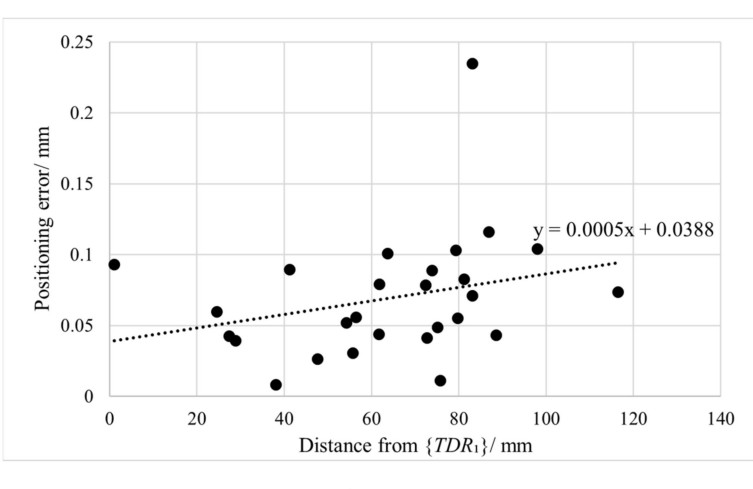

(**b**)

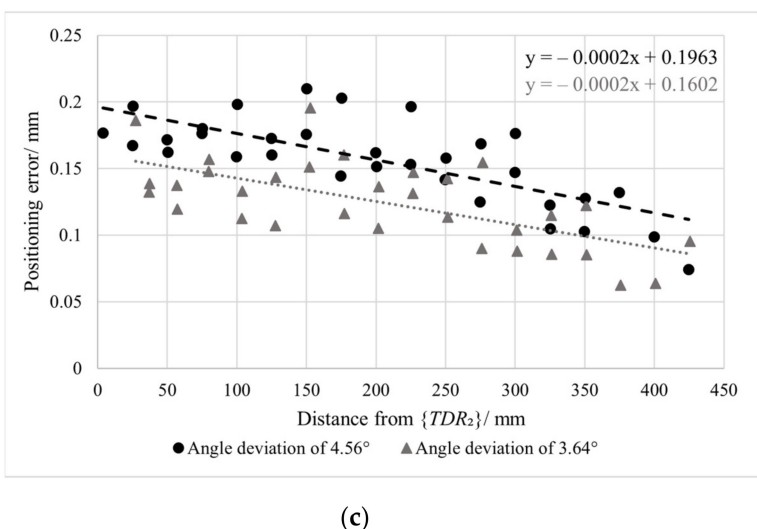

(**c**)

**Figure 11.** Spatial correlation analysis of residual error. (**a**) Positioning error of $\Omega_1$ vs. distance; (**b**) Positioning error of $\Omega_1$ vs. angle deviation; (**c**) Positioning error of $\Omega_2$ vs. distance and angle deviation.

The evidence from this experiment suggests that the calibration uncertainties of the TCP frame are the main factors affecting the positioning accuracy of the robot in the drilling

task of large orientation variation. Additionally, calibrating $^W T_T$ in different regions can make better use of the error similarity of the robot to reduce the error. What's more, it can also be identified that the method in Ref. [28] cannot eliminate the calibration error of the TCP frame. Although increasing the number of sampling positions with different orientations can improve the accuracy to a certain extent, the number of sampling points will increase dramatically. Therefore, by the regionalized calibration method proposed in this paper, the positioning accuracy of robots in $\Omega_1$, $\Omega_2$, $\Omega_3$ and $\Omega_4$ can fully meet the requirements of the drilling task, and the sampling time is also controlled within a minimum range, which provides a novel compensation method very suitable for robotic drilling task under large orientation variation.

## 5. Conclusions

This paper analyzed the influence of TCP calibration uncertainties on robot positioning error similarity. In an effort to eliminate the uncertainties, a novel TCP calibration method and an error compensation method using regionalized error similarity were proposed and adopted in the robotic machining system. Using these proposed methods, the positioning accuracy in the drilling task with a large orientation variation of TCP was improved to 0.23 mm, which meets the machining requirements of aerospace parts.

It is worth noting that the orientation variations in the study primarily occurred in the direction around the x-axis of the robot base frame due to the peculiarities of the parts being processed. For the drilling task with large orientation variation in the three directions, it becomes imperative to establish multiple TCP frames, which would also lower the efficiency. Further study is still needed to determine the division strategy of the workspace and its influence on positioning accuracy.

**Author Contributions:** Y.L.: Methodology, Investigation, Formal analysis, Writing—original draft. B.L.: Conceptualization, Funding acquisition, Writing—review & Editing. X.Z.: Resources, Supervision, Validation. S.C.: Supervision, Validation, Writing—review & editing. W.Z.: Software, Investigation, Formal analysis. W.T.: Project administration, Funding acquisition, Supervision. All authors have read and agreed to the published version of the manuscript.

**Funding:** This research was funded by the National Natural Science Foundation of China, Nos. 52075256, 52005254 and U22A20204.

**Institutional Review Board Statement:** Not applicable.

**Informed Consent Statement:** Not applicable.

**Data Availability Statement:** Not applicable.

**Acknowledgments:** This work has been co-supported by the National Natural Science Foundation of China (Nos. 52075256, 52005254 and U22A20204) and the Jiangsu Key Laboratory of Precision and Micro-Manufacturing Technology.

**Conflicts of Interest:** The authors declare no conflict of interest.

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
