# Peer review of "Error Similarity Analysis and Error Compensation of Industrial Robots with Uncertainties of TCP Calibration"

_applsci, doi:10.3390/app13042722_

Round 1

Reviewer 1 Report

The article proposes a strategy to compensate pose errors of the robot in machining process with uncertainties of TCP calibration. The contributions are to analyze the influence of TCP calibration uncertainties on error similarity, to develop a regionalized error similarity compensation method and to verify the proposed method by robotic drilling test. The article has a clear contribution, although it suffers from many stylistic and grammar deficiencies. The sentences are often too long, which makes the paper not very easy to follow. Additionally,  the notations are often confusing and lack clear explanation. In summary, the article presents a good research work, but it needs to improve presentation before the article is accepted for publishing.

Reviewer 2 Report

This paper presents a novel approach for improving the positioning accuracy of industrial robots used in aerospace manufacturing by addressing the issue of TCP calibration uncertainties. The paper proposes a TCP calibration method and an error compensation method that uses regionalized error similarity. The proposed methods were adopted on a robotic machining system and the results show that the positioning accuracy in a drilling task with large orientation variation of TCP was improved to 0.235 mm, which meets the machining requirements of aerospace parts.

It is acknowledged in the paper that the orientation variation in this study mainly exists in the direction around the X-axis of the robot base frame. However, it states that for drilling tasks with large orientation variations in three directions, it is necessary to establish multiple TCP frames, which would lower the efficiency. The paper also acknowledges that there is a need for further study on the division strategy of the workspace and its influence on positioning accuracy.

However, it is acknowledged that the method has limitations and further research is needed to improve the efficiency and generalize the method to other types of orientation variation. 

For example, the proposed method is only tested for orientation variation around the X-axis of the robot base frame. In order to generalize the method to other types of orientation variation, it would be beneficial to test the method for variations in other directions as well.

The paper also states that for drilling tasks with large orientation variations in three directions, it is necessary to establish multiple TCP frames, which would lower the efficiency. Further study could be done to explore a strategy to divide the workspace in a way that would minimize the number of TCP frames needed while maintaining good positioning accuracy. 

In the end, in the reviewer's opinion, in order to enhance the reliability of the proposed method, it would be good to see the results of more experiments with different types of materials, different shapes and sizes of objects, and different environmental conditions. 

Reviewer 3 Report

This manuscript presents a novel regionalized compensation method for improving the positioning accuracy of the robot with calibration uncertainties and large orientation variation of the TCP. It is a very interesting study, but the current version of this manuscript is not good enough to be published in Applied Sciences and my suggestion is ‘major revision’. I have the following comments. 

1 Authors points out in line 30 that the positioning accuracy of the robot is merely ±1 mm using traditional control methods. What traditional methods does the author refer to?

2 On line 71, the requirements of the aerospace manufacturing are 0.25mm. What is the basis of this opinion?

3 What are the criteria that define a large orientation variation and a small orientation variations?

4 There are many grammatical and spelling errors, for example, 'Zeng [16,17] and developed estimation methods...'.

5 Authors use the method in literature [17] as a comparative method, which should be carefully analyzed in the introduction.

6 In the introduction, authors point out that other methods were not accurate enough, but did not analyze the reasons. The author needs to emphasize the motivation for the new method and emphasize the characteristics of the new method.

7 What do the ‘Dilling poses’ mean in Figure 9 and a detailed analysis of Figure 9 is lacking.

8 The amount of data in the experiment is not enough to support the conclusion that the accuracy can reach 0.235mm.

Reviewer 4 Report

The submitted manuscript "Error similarity analysis and error compensation of industrial robots with uncertainties of TCP calibration" corresponds to the Journal's scope. The paper is devoted to the actual problem focused on the accuracy of industrial robots. Moreover, a novel TCP calibration method and an error compensation method using regionalized error similarity are proposed and adopted in the robotic machining system. It ensures the positioning accuracy in the drilling task with large orientation variation of TCP was improved.

The authors completed theoretical studies and performed simulation modeling and experimental studies. The paper is well-presented.

However, I can specify the following comments:

1.     Do not duplicate phrases/terms from the title in the keywords.

2.     It is unclear why the KUKA KR500-2830 industrial robot is selected. A brief justification is needed.

3.     How can a specific industrial robot's obtained results be transmitted to others?

4.     Please improve the Discussion. It must provide a comparative analysis of the developed techniques with similar solutions to other scientists.

Round 2

Reviewer 2 Report

The authors have addressed the issues, reviewer will accept the current format.

Author Response

We are pleased to know that our revised manuscript has effectively addressed your concerns. We appreciate your time and effort in reviewing our work and providing insightful feedback.
